# Don't Give Up on Democratizing AI for the Wrong Reasons

**Zimmermann, Annette**
Department of Philosophy
University of Wisconsin–Madison
zimmermann6@wisc.edu

**Zeppa, Andrew**
Department of Philosophy
University of Wisconsin–Madison
zeppa@wisc.edu

**Pandey, Srijan**
The Information School
University of Wisconsin–Madison
spandey43@wisc.edu

**Diao, Kenneth**
University of Wisconsin–Madison
kdiao2@wisc.edu

## Abstract

The claim that the AI community, or society at large, should 'democratize AI' has attracted considerable critical attention and controversy. Two core problems have arisen and remain unsolved: conceptual disagreement persists about what democratizing AI means; normative disagreement persists over whether democratizing AI is ethically and politically desirable. We identify eight common AI democratization traps: democratization-skeptical arguments that seem plausible at first glance, but turn out to be misconceptions. We develop arguments about how to resist each trap. We conclude that, while AI democratization may well have drawbacks, we should be cautious about dismissing AI democratization prematurely and for the wrong reasons. We offer a constructive roadmap for developing alternative conceptual and normative approaches to democratizing AI that successfully avoid the traps.

## 1   Introduction

Calls to "democratize AI" have become ubiquitous in academic and policy debates. Yet what democratizing AI means in concrete terms, which changes and interventions it would require, has remained analytically imprecise and marred by entrenched disagreement. While some promote democratization as an essential corrective to the concentration of power in AI industry, others—sometimes reasonably, sometimes not—express unease or outright opposition to the idea. Are these critics reacting to the right target? Is democratization of AI a misguided project, or merely a misunderstood one? We argue that recent debate on the democratization of AI has been characterized by two related yet distinct problems. First, discourse around democratizing AI is often shaped by inaccurate conceptions of what such a project would entail. For example, some interpret democratization as 'ordinary' citizens' mere ability to *access* and *use* cutting-edge AI tools, while others have in mind significantly more demanding conceptions of meaningful democratic control over AI design and deployment decisions [1]. Second, these confusions fuel skepticism. While we recognize that reasonable people will disagree about the relative value of democracy in comparison to other normative goals, as well as about the best means of realizing it, we argue that many critics risk dismissing democratization for the wrong reasons, and that many proponents risk failing to articulate its strongest version.

In our view, this double failure—conceptual and normative—has distorted the debate. It has led some to dismiss the ethical and political appeal of democratizing AI too hastily. It has led others to endorse it without sufficient clarity about its implications for citizens' and tech practitioners' personal responsibility as well as far-reaching policy change. Our aim is to resist both tendencies. We do not claim that democratizing AI is the only goal worth pursuing, nor that all interpretations of this goal that do not align with our preferred substantive account of democratizing AI are necessarily defective.

39th Conference on Neural Information Processing Systems (NeurIPS 2025) Position Paper Track.

Our contribution, as an interdisciplinary team rooted in philosophy, computer science, and science and technology studies, is twofold. First, we aim to offer a clearer map of the landscape: a survey of the major positions, tensions, and values at stake in the democratization debate. This map is designed to help others reason more clearly about what democratizing AI might *mean* and why it might *matter*. Second, we identify and endorse a specific substantive view—one that we think deserves more attention than it has received. That view is that we ought to democratize AI by harnessing the distinctive value of *non-technical expertise* held by 'regular' citizens for enhancing the quality of collective decisions on AI policy; by adopting a realistic view about the limits of democratization when it comes to securing *just* outcomes; and by resisting the assumption that democratization necessarily slows us down in an *objectionable* way. While a comprehensive *policy blueprint* and a comparative *empirical* investigation of different political strategies for democratization is beyond the scope of this *conceptual* and *normative* paper, we provide brief concrete examples of which levers of power are available to our audience of citizens, researchers, and policy experts.

In Section 2, we identify five common misconceptions—which we call 'traps' to acknowledge that these misconceptions are easy to mistakenly adopt—about the conceptual prong of the problem: the question of what democratizing AI means. Our ambition here is to equip our community to reason in more nuanced ways about their preferred views about democratizing AI, whether they are ultimately sympathetic or hostile to this goal. Having provided a clearer map of the conceptual terrain, in Section 3, we turn to the normative domain by critically comparing different views about whether democratizing AI would be ethically and politically desirable. Here, we analyze three common strategies for critiquing this goal, all of which in our view turn out to be traps: they are vulnerable to plausible objections, and while they may give us some reasons to approach the possible goal of democratizing AI with *initial* skepticism, they ultimately fail to *decisively* undermine the normative desirability of that goal. Further, we do not deny that yet *other* strategies for critiquing this goal, not covered here, may possibly be available, and that they may be less flawed than the three under discussion here: our aim is simply to tidy up the normative problem space, thereby ultimately improving even the quality of arguments and positions with which we disagree. In Section 4, we synthesize the normative implications of these points into recommendations for realizing our account of democratizing AI in practice. In Section 5, we discuss possible alternative views on democratizing AI beyond the traps identified in this paper, while explaining why we do not consider them to be offering decisive reasons to reject our preferred view.

## 2 Misconceptions about what democratizing AI means

### 2.1 The Democratic Minimalism Trap

While we grant that few would explicitly endorse a clearly reductive view of democracy like 'democracy is just about voting once every four years' or 'democracy is all about aggregating people's preferences', we nevertheless worry that an unduly narrow view of what successful democratization is or can be may still be an implicit background assumption driving increasingly popular views about democratizing AI. A common intuitive understanding of the democratization of AI is that it involves merely soliciting *input* from citizens or intervening in a *one-off* way along a single dimension—for example, by regulating some AI tool via a one-time ballot proposition or by restricting its deployment through state ballot measures [2]. Although these are certainly ways one might subject AI to democratic pressure, they are neither necessary nor sufficient for meaningful democratization. Individual interventions of this kind tend to produce desired outcomes at single incision points while leaving other upstream and downstream decisions just as open to unilateral, nondemocratic control. For example, despite laws limiting the use of AI facial recognition tools by law enforcement in various states and municipalities, police departments have pivoted to using tools like Track, a nonbiometric alternative to facial recognition that tracks people by other attributes such as body size, gender, hair color and style, clothing, and accessories [3]. By using alternative AI tools like Track, law enforcement agencies avoid breaking democratically imposed regulations while continuing to make decisions that arguably flout the spirit of those individual democratic interventions.

Additionally, some democratic procedures risk privileging one mode of consensus-building over others; thus systematically excluding marginalized voices and failing to create a truly open marketplace of ideas. For instance, a public forum may allow some citizens to contribute their unique perspectives to the conversation while inadvertently excluding others unable to participate due to financial, geographical, employment, or ability-related constraints. No single method of democratic

intervention, when implemented in isolation and without due concern for its societal context, is enough. Rather, to meaningfully democratize AI is to robustly expose it to the full range of the democratic process [4]. Concretely, this means that there ought to be multiple avenues for engagement, input, and participation [5], including but not limited to: informed public deliberation [6, 7, 8], the fair election of representatives, citizen assemblies and similar participatory models, continual audit and critique of decisions and decision-makers by third-party experts [9], direct agenda-setting and decision-making by citizen-participants, and the open access availability of information that is in the public interest [10]. We join a rich body of existing interdisciplinary work in highlighting that democratization is not a one-off intervention. Ideally, it is instead a multi-faceted, ongoing process, occurring within a complex system of participatory mechanisms and occurring at all stages of AI deployment: in order to realize this in practice, the different mechanisms above ought to be subject to *legislatively mandated iterations* at suitable intervals, such as annual audit and reporting obligations.

## 2.2 The Access Trap

Another possible misconception is that democratizing AI simply means increasing *access* by enabling 'ordinary citizens' to use AI tools, and by making models open-source, datasets publicly available, or platforms more transparent and user-friendly. An example of such a view is the claim that the "democratization of artificial intelligence means making AI available for all" [11]. We call this the Access Trap: this view implies an insufficiently demanding picture of democracy, one that fails to conceive of democracy as more than expanding access to a tool [12, 13, 14]. Under this view, democratization becomes a design problem—one solvable by technical tweaks rather than structural change [15, 16]. Yet access without agency is not empowerment [1, 15]. It is often a way of preempting and managing discontent without redistributing real control over AI design and deployment choices. For example, opening up an AI model for anyone to fine-tune or integrate into their product is seen as inherently empowering. To illustrate, consider Mark Zuckerberg's claim that "Open source [AI] will ensure that more people around the world have access to the benefits and opportunities of AI, that power isn't concentrated in the hands of a small number of companies, and that the technology can be deployed more evenly and safely across society" [17]. Unsurprisingly, though, the move to enable maximally widespread AI access is not purely (or even primarily) altruistic: "We [Meta] benefited from the ecosystem's innovations by open sourcing leading tools like PyTorch, React, and many more tools. This approach has consistently worked for us" [17]. In this way, access can fail to alleviate or even deepen inequality generated by concentrations of economic and political power in Big Tech when offered without democracy-promoting structural change [18, 19, 20].

We encourage the AI community to avoid the Access Trap and to reject the idea that technical openness is sufficient [21]. Democratization ought to be measured not by how many people can use a tool, but by how many people have meaningful control over its existence, direction, and impact. This requires shifting focus from access to decision power and control over resources. Not all communities want access to harmful systems; many want the ability to say no to their deployment altogether. Hence, true democratization must include the right to refuse, to redirect resources, and to imagine alternatives [22, 23]. Ideally, AI democratization captures a fuller spectrum of possible views and lived experiences in society, so that systems can be responsive to those perspectives rather than merely inviting in the latter once key decisions have already been made [13, 24]. Here, we see at least two possible concrete mechanisms: first, *resource reallocation* in the form of fines (e.g. Italy's privacy watchdog's 2024 decision to fine OpenAI €15 million over GDPR violations) or direct compensation of citizens for technology-related harms (e.g. the 2022 TikTok $92 million class action settlement for US users) and second, bottom-up civic rather than top-down decision-making about *public AI funding*, by harnessing innovative democratic mechanisms such as ballot propositions or citizen assemblies that have proven successful in related policy domains for the specific purpose of making decisions about publicly funding AI research and deployment.

## 2.3 The Preference Satisfaction Trap

A popular, though not uncontroversial, view holds that the core aim of democracy is to satisfy (rather than actively aggregate) citizens' preferences. From this perspective, the mechanisms by which those preferences are identified or acted upon—whether through voting, deliberation, or algorithmic inference—are secondary. What matters is preference alignment, not actual participation. We appreciate that this line of reasoning has practical appeal at first blush. If public preferences can be measured, modeled, or predicted, democratic legitimacy might seem to require no more than

sufficiently responsive AI policy: governments and corporations should simply do what people want or what is in their best interests. Against this, we insist that democracy without actual participation misses democracy's essential constitutive feature [22, 25]. Thus, we deny that hypothetical participation in a mere 'preference satisfaction' sense can generate real democratic legitimacy.

There is, however, a less strong, less obviously flawed alternative version of this view, which merits further discussion. Supporters of a preference satisfaction account of democracy may think that if actual participation by the entire democratic constituency is difficult to realize in practice, then perhaps using the input of a small group of citizens to build a model of a larger constituency is the best available path for overcoming this feasibility constraint. We classify well-known approaches like Anthropic's Collective Constitutional AI [26] under this umbrella, though to be charitable to this approach, we acknowledge that these authors do not endorse the implausible claim that their approach would be sufficient for *fully* democratizing AI. Nonetheless, even this version of the view risks being reductive in a way that resembles the traps covered already. It fails to insist on a sufficiently tight link between the concept of democratic legitimacy and the process of *actually* voicing one's normative views, of *actually* contesting competing values amongst our fellow citizens [27], and of *actually* demanding accountability from our elected representatives. If we reduce democracy to preference satisfaction based on the alignment of AI tools with nonbinding citizen input, we miss what makes democratic procedures distinctive and legitimate: the idea that people ought to have collectively binding power to determine the conditions under which they live, not only as individuals with self-regarding interests, but as active members of a shared, reciprocal political project.

## 2.4  The AI Exceptionalism Trap

The Exceptionalism Trap is the misconception that democratizing AI involves unique, outsized, indeterminate, or otherwise special challenges that make it exceptionally different from the task of democratizing other areas of science and technology [28, 29, 30]. The line of reasoning is roughly as follows: (1) AI is exceptionally different from other areas of science and technology that we might want to democratize. (2) Therefore, the task of democratizing AI is itself exceptionally different from the task of democratizing other areas of science and technology. Often, these ideas are implicit in arguments about the practical infeasibility of creating the public institutions needed to democratize AI, or about corporate actors' ostensibly better positioning as stewards for AI in comparison to 'regular' citizens and public officials. The Exceptionalist Trap is understandable given AI's novelty and power. Indeed, issues like opacity and explainability, among others, do perhaps pose genuinely unique challenges for responsible AI deployment [31]. But even if one grants that claim (1) is true—and there are good reasons to doubt this [32]—claim (2) does not follow.

Indeed, many of the challenges that might be thought particular to the democratization of AI are likewise challenges found in a wide variety of scientific and technological endeavors [33]. For example, the problem of reconciling a "thick" (i.e., jointly descriptive and normative) and controversial concept like fairness with particular formal statistical measures is one that a democratic task force might face given the responsibility of regulating AI deployment in criminal-justice contexts [34]. But measurement processes in general have long been recognized by philosophers of science as involving background assumptions about the relevance and correctness of particular theories [35, 36, 37], which in turn rely on non-epistemic value judgments for their development and selection [38, 39]. Thus, the theory-ladenness and consequent value-ladenness of measurement practices extends well beyond such obviously thick concepts like fairness. For example, concepts such as "biodiversity" [40, 41] and "well-being" [42, 43] face similar difficulties, given the plurality of normative judgments that might enter into their operationalization. Democratization of measurement is just as much a challenge in these domains as it is for AI [44]. In general, the challenges of democratizing AI are not necessarily different in kind from the challenges of democratizing other areas of science and technology, though they may well be different in terms of urgency and scale.

If there are higher stakes associated with developing and deploying AI because of the special technical and societal challenges it presents, democratization becomes all the more important. For instance, to democratize the measurement of thick concepts like algorithmic fairness, the relevant value-judgments should ideally be made through a legitimate and representative political process rather than by a small group of unelected corporate decision-makers. Furthermore, because specific affected groups will have better firsthand access to the normatively relevant details in any given measurement context,

such groups are uniquely positioned to help direct the integration of the broader democratically informed values into the technical treatment of the measurand [45, 46, 47, 48].

## 2.5 The Egalitarian Illusion Trap

The Egalitarian Illusion Trap rests on the incorrect assumption that democratic procedures, once established, *necessarily* create political equality, defined as the status of each member of the democratic constituency as a public equal endowed with an equal cluster of participation rights and opportunities, as well as correlative ethical and political obligations [49, 50, 51]. The Egalitarian Illusion Trap assumes that if everyone can participate in consultations or voting processes, then all participants have equal power to influence outcomes. It overlooks the structural inequalities that affect whose input is actually heard, trusted, and acted upon [15, 52].

Wide acceptance of the view underpinning the trap could pose a threat to democratic institutions by enabling antidemocratic political behavior by bad actors: it could allow groups or institutions to claim democratic accountability while actively sustaining unequal power relations in society [53]. This reasoning trap ignores the basic fact that participation occurs under unequal social conditions. People come to democratic processes with radically different access to time, resources, language, and legitimacy. Some arrive backed by institutions and disciplinary authority. Others arrive burdened by structural neglect and historical silencing. The Egalitarian Illusion trap is especially ethically acute when marginalized communities are asked to provide feedback on already-deployed AI tools without being consulted on policy or infrastructure matters [52]. Systems are often trained on data extracted from marginalized communities—data collected without meaningful consent, from lives shaped by systemic inequality [54, 55, 56]. These systems do not become ethically innocuous simply because they are made transparent or open to feedback: the invitation to speak, on terms set by corporations and top-down political institutions, is not the same as the power to decide [57, 58].

Democratic processes can be harmful when used to confer legitimacy on outcomes that entrench and simultaneously obscure existing hierarchies[59]. Our concern here mirrors recent critical work on checklist-style approaches to fairness audits and AI ethics frameworks, which highlights how such approaches turn rigorous ethical deliberation into superficial compliance exercises only loosely informed by token public input [60, 61, 62, 63]. These practices calcify institutions by preempting demands for deeper structural transformation: this literature calls this phenomenon *ethics-washing* [52, 53, 64]. We see a parallel problem possibly arising for AI democratization: *democracy-washing* [65]. To prevent the latter, and to resist the Egalitarian Illusion trap, one or ideally several levers of power in a democratic constituency—public officials, researchers, citizens, and (to the extent that restrictive market incentives allow this) tech workers and industry decision-makers—must remain vigilant about what democratization can and cannot achieve. While *measuring* successful democratization, as opposed to democracy-washing, requires comparative empirical methods beyond the scope of this paper, our theoretical work does imply a coarse-grained acceptability threshold: democratization is insufficient if one actor (e.g. a tech company or bureaucracy) can unilaterally decide when and how to consult others, and can disregard those inputs at will.

# 3 Misconceptions about which arguments offer decisive reasons against democratizing AI

We now survey three strategies commonly pursued by those skeptical of the claim that democratizing AI is a desirable goal. In our view, none succeed, hence we classify them as traps; though this does not imply that *no* plausible objections against democratizing AI exist. Our view is thus in principle compatible with the claim that the goal of democratizing AI might be overridden by some other normative goal.

## 3.1 The Stupid Democracy Trap

Democratizing a policy domain often entails a real or perceived shift in the extent to which expert input influences decision-making. One common concern about introducing a broader set of voices into the decision-making process is that this redistribution of influence may diminish the outcome quality of decisions: if AI policy becomes less expert-driven, it may also become less informed and less effective [66]. We agree that many policy domains—including AI—plausibly depend to

some extent on forms of knowledge that are complex and technical. We also agree that low-quality, uninformed AI policy decisions risk causing significant harm: we see why one may wish to resist AI democratization on the grounds that democratic constituencies as a whole may make suboptimal decisions due to a lack of relevant AI literacy and expertise. However, the underlying assumption that only formally trained technical experts possess relevant insight is both empirically and normatively questionable: while this concern plausibly identifies potential prima facie *obstacles* to successful democratization, it fails to offer *decisive* reasons against the goal of democratization proper.

AI expertise is not the sole province of academics or technical professionals [1, 58, 66, 67]. Citizens often have forms of local, experiential knowledge that are not entirely accessible to government bureaucrats, industry practitioners and academics, but that are nonetheless indispensable to good collective decision-making. These forms of knowledge are especially important in areas where policy choices have context-specific effects. Citizen input can thus have distinctive *epistemic value* in comparison to other kinds of input: citizens can supply insights that plausibly count as domain-specific expertise in their own right, even if that input is less technical than the input of other experts. Citizen input is *particularly* valuable when decisions about AI turn on normative, not technical trade-offs (e.g. freedom vs. equality, efficiency vs. transparency).

In a nutshell, there are good reasons to resist the claim that democratic decisions are 'stupid' in a way that decisions made by a smaller group of experts are not: citizen participants can serve as *representative experts*. Technical specialists often operate within narrow institutional cultures, which are not always aligned with the values and needs of the broader population [18]. Citizen participation enhances the quality of decisions by helping democratic constituencies make salient different normative views about AI policy questions held by different groups in society, thus boosting decision quality rather than undermining it [44]. Sortition mechanisms like the ones underlying citizen assemblies serve the goal of widening representation particularly well. For a concrete example of the epistemic value of citizen representative experts, a recent citizen assembly on AI and freedom in Germany surfaced value-based preferences among citizens that were noticeably different from current German top-down AI policy, and called explicitly for a permanent establishment of stronger participatory mechanisms: these preferences would not have been salient without this process [68].

Relatedly, citizens may serve as *representative critics* who productively raise concerns that might otherwise go unvoiced [69]. Whereas expert communities often converge on shared views and priorities, and whereas even strongly public interest-minded industry and policy experts may be *de facto* incentivized to prioritize values that do not closely track the actual value preferences of the larger constituency, 'regular' citizens may offer a useful counterweight. For example, suppose that industry professionals and public officials in a given state converge on the judgment that AI ought to be deployed to render some process (say, policing) more efficient: here, the wider population may endorse values that constructively push against prioritizing the value of efficiency over other values (e.g. fairness, accountability) in this domain. Consider the case of bottom-up civic activism contributing significantly to San Francisco's *2019 Stop Secret Surveillance Ordinance*, and consider the decision of multiple technology companies, in response to citizen activism bolstered by influential critical scholarship, to temporarily stop deploying facial recognition tools in policing [70, 71]. For another concrete example of successful citizen participation, consider the 2023 Writers' Guild of America strike due to film studios' use of AI coupled with complaints about low compensation, which culminated in an agreement between WGA and the Alliance of Motion Picture and Television Producers that placed restrictions on AI use. Such cases show that democratic input by citizens acting as *representative critics* may yield higher-quality, more representative AI policy.

None of this implies that democratic participation replaces technical expertise. The challenge is to design institutions where expert and citizen input are in dialogue rather than competition. Our view avoids the 'Stupid Democracy' Trap by fostering democratic practices that treat citizens not as tech-illiterate audiences but as co-participants with their own relevant expertise, while still recognizing the epistemic value of deep technical knowledge.

## 3.2 The Unjust Democracy Trap

AI democratization skeptics may be tempted to invoke a long-standing *general* problem of democratization, i.e. one not specific to *AI* democratization: the democratic majority may overlook or even deliberately override the interests and preferences of democratic minorities. This is a familiar and philosophically well-understood problem that democracies must grapple with more generally. As

important as this known problem continues to be, it does not constitute a sufficient reason to reject the specific goal of democratizing AI, both because (i) there are existing institutional guardrails to help mitigate the problem and because (ii) there are other non-instrumental reasons to keep pursuing democratization despite the presence of obstacles. For instance, with respect to the former, constitutional protections and anti-discrimination laws help protect the interests and rights of democratic minorities, imposing what political philosophers call 'minimum outcome constraints' on democratic processes, thus preventing those processes from infringing on individual rights and from veering too far from the normative goal of treating the morally weighty interests of all citizens with equal respect [25]. Regarding the latter, free and fair democratic processes are goods in themselves, separate from the desirable outcomes they produce. Democracy realizes in practice the normative values of autonomy and collective self-determination [72], and thus tracks the morally weighty interests of human beings that many philosophers consider to be truly *fundamental*. One possible intuition underpinning this idea, which we think is reasonable, is that setbacks to this core interest in autonomy would also set back *other* important morally weighty interests that people tend to pursue by *exercising* their autonomy, ideally as part of procedures that treat them as free equals, such as safety and welfare.

In sum, rejecting AI democratization on grounds of democratic processes' potential to overrule democratic minorities not only ignores available mitigation techniques, but also fails to consider the intrinsic value of autonomy, put in practice by the idea of letting people determine their own lives via the implementation of fair democratic procedures for all members of the constituency [25]. We do not deny that there may be scenarios in which countervailing normative considerations *outweigh* the moral weight of democratic decision-making's intrinsic value: for instance, when an imminent disaster threatens large-scale harm, giving plausible reasons to *temporarily* prioritize public safety, harm reduction, and welfare over autonomy realized through fair democratic procedures. A full comparative ranking of such values across all possible cases, and a comprehensive philosophical defense thereof, lies beyond this paper's scope. We instead offer a modest, ecumenical claim: the moral and political goods of democratization should generally be ranked highly *by default*, placing the burden of proof on those invoking weightier, overriding values to justify temporarily deprioritizing democratization.

A related potential objection to democratizing AI focuses on the risk that once an institution endorses the language and aesthetics of democratic participation, this may have a *de facto* chilling effect in that collective decisions become insulated from critique [12, 73]. The concern is that democratizing AI might bring about unjust outcomes in light of the Egalitarian Illusion Trap introduced above. For instance, bad or misguided actors in the public and private sector may gather consultation data, convene community panels, and solicit feedback, not to equally distribute decision power, but to consolidate it [15, 74, 75]. Once such processes occur, the resulting systems may be branded as publicly governed or community-informed, even when they continue to reflect the interests of dominant actors [1, 12, 15]. Superficial democratic mechanisms can thus, counterintuitively, become techniques of depoliticization, undermining citizens' rights to contest AI policy outcomes by positioning dissent as already accounted for.

Crucially, however, neither our point that democratization can in some cases yield unjust outcomes, nor our point that democratic processes can be misused for the purpose of democracy-washing those outcomes, should be taken to imply that we should reject the normative goal of democratization itself. The problems we discuss here do not indicate a failure of democracy as a political project, but a failure of doing democracy right [73]. We worry that positions that reject democratic processes purely on the grounds that they could create and legitimize unjust outcomes imply that the alternative must lie elsewhere outside of the democratic domain entirely, perhaps in technocratic governance or (ostensibly) benevolent corporate stewardship. Both would undermine key values characterizing a just society more so than the democratic alternative would, as unlike these alternatives, democratic governance offers the possibility of productive contestation and of publicly accountable self-correction.

### 3.3 The Slow Democracy Trap

A recurring objection to democratizing AI is that democratic processes are inherently too slow for the pace at which technology evolves. Critics argue that public deliberation cannot keep up with the pace of global competition in a capitalist market [for a canonical statement of this view, see [76]]. Within this framing, democracy is cast as a procedural burden that obstructs innovation and progress. This argument presents speed as inherently virtuous and slowness as a vice. It fails to

recognize that the current accelerated pace of AI development is not an accident, but a result of institutional choices. When speed becomes the overriding value, it crowds out democratic judgment and forecloses opportunities for public contestation.

One way a speed-first approach might be thought justifiable is as a response to the fear that AI is at the center of a contemporary arms race [for representative recent statements of this widely shared view, see [77]]. This fear frames the rapid development of AI as a matter of national security, or even as a matter of securing the future of Western democracies against the would-be dominance of competing ideologies [78]. For instance, a recent advertisement for Palantir Technologies warns that "A moment of reckoning has arrived for the West" [79]. Although general concerns about geopolitical competition are not unfounded, the version of this worry that leads inevitably to the conclusion that speed must be prioritized above all else assumes a false dichotomy—either develop as rapidly as possible or become vulnerable to the overwhelming dominance of a competing global power [80]. But this dichotomy overlooks the possibility of transnational cooperation, which is a way forward that not only has historical precedence, but is also likely mutually appealing for all parties, given the broader destabilizing potential of AI [81].

Moreover, democratic slowness is not *objectionable* in all cases. It can be a critical feature of good governance that takes AI risks seriously. It allows time for dissent to emerge and for decisions to be redirected. For communities who are historically excluded or directly harmed by AI systems, this slowness could function as a productive political safeguard. It hinders the unchecked deployment of technologies that often deepen inequality and intensify surveillance, thereby possibly undermining key rights of democratic citizens. We grant that democratization may slow the process, but not always objectionably so, given that slowness could bring about an overall better, more legitimate outcome.

## 4 Recommendations

(1) **Take seriously an appropriately demanding and conceptually accurate view of democracy.** As we have argued throughout Sections 2.1-2.5, democracy is not reducible to simple access, mere consultation, or the aggregation and satisfaction of preferences. Nor is it exhausted by episodic voting or surface-level input. Ideally, democracy aspires to enable meaningful and ongoing participation opportunities for all citizens to exert real influence over collective decision-making. To democratize AI, we must design institutions that enable civic control over how AI systems are developed and deployed, even—as argued in 3.3—if this requires additional time investment. This entails not only inclusive procedural mechanisms, but also broader safeguards against the political capture of decision-making processes by powerful firms or unaccountable political actors.

(2) **Do not place unrealistic expectations on democratization.** Our arguments in Sections 2.5 and 3.2 jointly imply this: democracy is valuable, but it is not a universal solution to all normative or institutional problems. Democratic processes do not guarantee political equality, substantively just outcomes, and high-quality, rational decisions. By themselves, they cannot resolve problems of geopolitical competition and oligopolistic wealth concentration. Nor do they prevent polarization, misinformation, or antidemocratic social movements. By recognizing these limitations, we do not diminish the value of democratic ideals, and we do not view them as compelling reasons to abandon the goal of democratization. Defending AI democratization requires an even-handed account of what democracy can deliver in this domain—and a recognition of what lies beyond its reach.

(3) **Resist exceptionalist thinking about AI.** Our arguments in Sections 2.4 and 3.1 imply this recommendation. It is tempting to see AI as uniquely resistant to democratic control—too fast, too global, or too complex to be shaped by lay input. Whenever a domain becomes technically specialized and economically powerful, arguments arise that it must be insulated from democratic interference. But institutional challenges raised by AI—information asymmetries, concentrated power, cross-jurisdictional spillover effects—have precedents in other policy areas: climate, public health, finance. We agree with an emerging 'debunking' school of thought, one skeptical of AI's alleged special status in recent literature (notably [32]), which in our view implies that AI is not uniquely exempt from democratic governance. The key question is not *if* AI should be democratized, but *how* institutional design changes can make that goal achievable.

(4) **Treat AI democratization as a weighty, but defeasible, normative goal.** We hold that the project of democratizing AI is worth pursuing as part of a broader substantive philosophical account of democracy defined as a political ideal that optimizes for actual participation and for heterogeneous,

domain-specific expertise, including—as argued in 3.1—non-technical expertise specific to 'regular' citizens. When applied to the governance of AI systems, this ideal supports efforts to build participatory processes that counter technocratic or corporate control that undermine democratic legitimacy. But we have been clear, as per Section 3.2, that even if one agrees with our conclusions, it does not follow from our view that democratization should override all other values in all circumstances. Defending the value of democratization is compatible with recognizing its context-specific limits.

## 5    Other Alternative Views

We have included our rebuttal of each of the alternative views that lead to different traps in sections 2 and 3. But there are yet other alternative views beyond those traps. For example, in this paper we have not focused in depth on civic tech approaches, which often overlap with views supporting AI democratization, but are nonetheless distinct from them. Our view of AI democratization is about increasing democratic control over AI design and deployment decisions; to use democratic mechanisms to improve AI and to enhance the legitimacy of AI policy. By contrast, civic tech approaches aim to use AI directly to optimize democratic procedures and institutions, for instance by enabling better public deliberation or by analyzing large sets of political preferences [82, 83, 84]. These are potentially useful approaches, in principle compatible with our position. As such, there is no burden on our account to offer a rebuttal. However, we caution the community that some versions of civic tech approaches may collapse into tech solutionist, paternalistic, non-participatory preference satisfaction approaches that we critique above. We are particularly concerned about civic tech initiatives that center around providing access to AI for problem-solving—e.g. policing, housing, education—without interrogating underlying unjust social structures [14, 85, 86]. Communities already harmed by data extraction and surveillance may be invited to "participate" only after the system has been built, with no real opportunity to shape or challenge its direction. A notable example is the New York City Automated Decision Systems (ADS) Task Force, where community advocates criticized the lack of transparency and the exclusionary structure of public engagement [87]. As we have argued above, participatory approaches can make AI *appear* aligned with democratic values while replicating the power asymmetries they claim to fix [61, 64].

Due to space constraints we have also not engaged with views on AI governance that are neither pro-democratic nor anti-democratic, but rather what one might call '*ademocratic*': views to the effect that democracy is simply not a *relevant* value for a domain like AI. Such views raise philosophically complex questions about which societal domains are appropriately understood as *political* [88, 89]. They would thus require much more in-depth examination. For the same reason, we have also not defended a view about whether our position implies that *nondemocratic* societies ought to be democratized, or whether it merely implies, more modestly, that societies in which (possibly imperfect) democratic procedures are *already* in place ought to strive to enable greater democratic control over AI. We recommend further research and constructive academic dialogue in this area.

## 6    Conclusion

The idea of democratizing AI has gained salience, but its scope and normative implications remain unsettled. We have argued that this idea merits more conceptually clear examination before reaching conclusions—whether in favor or against. We invite the academic community to double down on engaging more closely with this topic. Doing so will include a pragmatic assessment of complex feasibility and implementation challenges in this area—*empirical* topics largely beyond the scope of this paper—as well as good-faith engagement with persistent moral and political disagreement over which goals the AI community *ought* to prioritize. Our arguments should not be taken to imply that democratizing AI will resolve all of its political and moral challenges, nor that democratization should always override other values. In many cases, the pursuit of democratic aims may need to be balanced against countervailing considerations. A serious account of AI democratization must make space for those potential normative tensions, as does our account. By developing these arguments, we have advanced a debate that is already justifiably attracting wide interest, but that remains frequently misunderstood and philosophically underexplored. We hope to have offered a clearer picture of what democratizing AI could mean and why it might matter—one that is sensitive to the complexities of the technologies involved and the normative landscape in which they are embedded.

## 7 Acknowledgments

Zimmermann gratefully acknowledges research support by the UW-Madison Vilas Faculty Early Career Investigator Award, the UW-Madison Fall Research Competition, and a Research Fellowship at the Weizenbaum Institute.

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
