# OpenReview forum: "Don’t Give Up on Democratizing AI for the Wrong Reasons"
_NeurIPS.cc/2025/Position_Paper_Track — NeurIPS 2025 Position Paper Track_

### Official Review · Reviewer_gNzk · 2025-08-04

**Significance:** 3
**Presentation:** 3
**Rating:** 7
**Confidence:** 3

**Summary:**

This position paper argues that the AI community should not abandon the goal of democratizing AI based on conceptual confusion or weak objections. It first clarifies what “democratizing AI” means, countering five common misconceptions that hinder meaningful discourse. It then critically examines normative objections to democratization—such as it being unrealistic, dangerous, or ill-defined—and identifies these as flawed or misleading. The authors argue instead for treating democratization as a complex but worthwhile normative aim. The paper concludes with a set of conceptual recommendations, encouraging a serious commitment to democratizing AI grounded in accurate understandings of democracy and the limitations of AI systems.

**Strengths:**

The paper makes a valuable contribution by unpacking and correcting misconceptions around what “democratizing AI” entails.
The authors take seriously both proponents and critics of democratization. They don’t dismiss opposition but instead clarify what the real disagreements are.

**Weaknesses:**

While the conceptual recommendations are sound, they are not accompanied by specific strategies for how democratization might be achieved in practice.

**Questions:**

Who is the intended audience for the recommendations (researchers, policymakers, industry)? What levers do they realistically have to act on this position?

**Alternative Position:**

Yes, and alternative positions are well-considered and addressed by the argument

**Author Identification:**

No.

**Context:**

3

**Discussion:**

3

**Ethics:**

["NO or VERY MINOR ethics concerns only"]

**Position:**

Yes, the paper argues for or against a position related to machine learning.

**Support:**

4

**Thoroughness:**

3

---

### Official Review · Reviewer_9rTh · 2025-08-08

**Significance:** 2
**Presentation:** 2
**Rating:** 6
**Confidence:** 4

**Summary:**

This paper critiques prevailing narratives around democratizing AI, arguing that both conceptual and normative misconceptions have distorted the debate. It defines democratization not merely as wider access to AI technology but as enabling meaningful public control over its development and deployment, including the right to redirect resources and refuse adoption. The authors reject AI exceptionalism and the democracy trap, which undervalues citizen expertise and participation. They propose that citizens are not just passive beneficiaries or testers but holders of crucial context-specific and representative expertise. The paper advocates for institutional design changes to promote participatory governance of AI while recognizing both the potential and the limitations of democratic processes in shaping technical systems.

**Strengths:**

The paper provides a sharp and timely critique of superficial uses of democratizing AI. It reframes the conversation in terms of institutional power, rights to refusal, and resource allocation, highlighting dimensions often ignored in technical discourse. The authors make a strong case against AI exceptionalism and emphasize the value of lay expertise, particularly from marginalized or non-technical communities. Their recommendations for participatory institutional design are both normative and actionable, expanding the discussion from access to agency.

**Weaknesses:**

The paper could benefit from clearer structuring of key arguments. The discussion of "traps" (e.g., stupid democracy trap, unjust democracy trap) feels conceptually rich but under-defined in parts. While the paper rightly critiques AI exceptionalism, it risks underestimating the real technical and infrastructural barriers to broad participation. More concrete pathways for integrating citizen influence into real-world AI governance structures (e.g., funding mechanisms, deliberative processes) would strengthen the practical impact of the proposals.

**Questions:**

What concrete mechanism or institutional structures do the authors envision for enabling meaningful citizen redirection of AI resources?

**Alternative Position:**

Yes, and alternative positions are well-considered and addressed by the argument

**Author Identification:**

No.

**Context:**

3

**Discussion:**

3

**Ethics:**

["NO or VERY MINOR ethics concerns only"]

**Position:**

Yes, the paper argues for or against a position related to machine learning.

**Support:**

2

**Thoroughness:**

5

---

### Official Review · Reviewer_T2BV · 2025-08-09

**Significance:** 2
**Presentation:** 4
**Rating:** 7
**Confidence:** 3

**Summary:**

The paper, "Don't Give Up on Democratizing AI for the Wrong Reasons," explores the concept of democratizing AI, arguing that the debate is often distorted by two problems: conceptual disagreement about what it means and normative disagreement about whether it's desirable. The authors identify eight "traps" or common misconceptions related to these problems. The paper advocates for a particular view of AI democratization that involves harnessing the non-technical expertise of regular citizens for policy decisions, being realistic about the limits of democratization for securing just outcomes, and resisting the assumption that it inherently slows down progress.

**Strengths:**

In my opinion, the main strength of the paper is that it is well-written and well-structured. The authors effectively use the concept of "traps" to categorize and address common misconceptions and critiques of AI democratization, providing a helpful framework for the reader and a useful reference/source for the AI community.

**Weaknesses:**

The paper does a good job at explaining that agency, structural change and input in the development and deployment of AI is desirable. As I see it, the main core recommendations of the paper are 1 - "we must design institutions that enable and promote participatory influence over how AI systems are developed and deployed" and 2 - "to build participatory processes and to prevent forms of technocratic or corporate control that undermine democratic legitimacy." That is fine. It would have been nice to see these recommendations supported by more evidence of where they were useful in the past.
Also, some points could be further elaborated and supported with evidence such as the input from regular citizens being valuable.
Please consider checking/fixing the double-quotes on line 101.
Section 4 recaps some concepts discussed above, which is good, but I felt too much of the paper was used to develop arguments about how to resist each trap and too little to defend the authors' recommendations.

**Questions:**

- The paper claims to provide a "constructive roadmap" for democratizing AI. Could you provide more specific, actionable examples of institutional designs and participatory processes that you recommend?
- You also mention that "democratization is not a one-off intervention". Could you elaborate on how you recommend that this continuous process should take place for the institutional designs and participatory processes you suggest?
- Lastly, are there any existing institutional designs or participatory processes that you believe are doing a good job?

A bit out of the scope of the paper, but perhaps still good to consider:
- What specific metrics or frameworks do you believe can be used to measure whether a democratization effort is genuinely redistributing power, rather than merely "democracy-washing"?

**Alternative Position:**

Yes, and alternative positions are well-considered and addressed by the argument

**Author Identification:**

No.

**Context:**

3

**Discussion:**

3

**Ethics:**

["NO or VERY MINOR ethics concerns only"]

**Position:**

Yes, the paper argues for or against a position related to machine learning.

**Support:**

2

**Thoroughness:**

3

---

### Note · Authors · 2025-09-05

**1-11 Submit Again:**

Definitely yes

**1-1 Submission Process:**

5

**1-2 Next Year:**

Continue offering a position paper track. It's a useful, much-needed format for the community and the process this year was well-organized. We are very pleased with the high-quality, remarkably constructive reviews we have received!

**1-3 Future Development:**

We don't have any suggestions for improvement right now. The paper length is good, the rigorous review process is good, the author instructions are good.

**1-4 Interest:**

["Panel discussions with other position paper authors", "Structured debates on controversial topics", "Workshops for developing position papers", "Mentorship programs for early-career researchers"]

**1-5 Thoughtful:**

9

**1-6 Supportive:**

9

**1-7 Technical Aspects Versus Position:**

9

**1-8 Gate Keeping:**

10

**1-9 Camera Ready Changes:**

We are delighted and immensely grateful to our three reviewers for their generous, supportive, and illuminating feedback.

All three reviewers agreed that it would be useful to include more concrete, actionable examples of when civic participation in AI policy was useful in the past and how these concrete strategies might be implemented going forward despite technical and infrastructural barriers. While a full taxonomy and empirical comparative evaluation of the full range of such strategies would be beyond the scope of this paper, we now offer more concrete examples (including bottom-up protests, citizen assemblies, consultative/deliberative models, third-party auditing) to illustrate more tangibly what increased AI democratization might look like.

Furthermore, in response to R1’s feedback, in our revised paper version we have explained more clearly in what sense civic input would be epistemically *valuable* in comparison to other kinds of input, we have clarified the logical connection between the ‘trap avoidance’ section of the paper and the ‘recommendations’ section, and we have explained how to make different forms of participation iterative.

In response to R2, we have enhanced the signposting of the end of section 1 and parts of 3. We also clarify the specific issue of how one could implement our point about redirecting *resources* to citizens in real political practice.

In response to R3, we endorse R3’s helpful ‘levers of power’ framing to better systematize the concrete examples of democratization that we mention in the paper.

**3-1 Review Response1:**

T2BV

**3-2 Reaction To Review1:**

1.R1 asked for more evidence of how participatory processes were useful in the past and which existing institutional designs are good. We now include 2 examples of civic bottom-up protests against AI deployment that led to tangible policy change, plus other participatory processes (citizen assemblies, 3rd party auditing).

2.R1 suggested clarifying where input from regular citizens is valuable. We explain that citizens contribute non-technical, value-based expertise.This input is particularly valuable when decisions about AI turn on normative, not technical trade-offs (e.g. freedom vs. equality).But we also clarify there are cases in which deep technical knowledge *is* essential. Our view is compatible with this: technical experts should in such cases lead (not replace) democratic deliberations.

3.R1"felt too much of the paper [focused on] how to resist traps and too little to defend the recommendations” We now include a sentence per rec to explain why the arguments in #2&3 logically imply the conclusions of #4.

4.R1 asked us to elaborate on“democratization is not a one-off intervention.” We originally cited political science work on different kinds of iterative participation (lines 88-93);we now explain *how* processes can be made iterative, e.g. by legislatively mandating regular input/audit intervals.

5.R1, stating“the paper claims to provide a 'constructive roadmap' for democratizing AI", requested more actionable examples. Building on this comment and R2’s positive feedback that our points are “normative and actionable,” we added such examples. (At the same time, we clarify our goal is not to provide a full “roadmap for democratizing AI.” Rather our abstract states we aim to provide a constructive roadmap *for developing alternative conceptual & normative approaches* to democratizing AI that avoid common traps:a map for reasoning rather than political preferences.Given limited scope, we cannot resolve empirical debates concerning a full political strategy.)

**3-3 Review Response2:**

9rTh

**3-4 Reaction To Review2:**

1.R2 notes that “the paper could benefit from clearer structuring of key arguments. The discussion of 'traps' feels conceptually rich but under-defined.” We appreciate this and, in response, added explicit signposting at the end of section 1 to explain why we chose these eight traps—namely, that they are the most commonly articulated views in the community. We also improved internal signposting in section 3 (stupid democracy trap, unjust democracy trap) to enhance clarity locally. We agree that these adjustments improve readability.

2.R2 worries that the paper underestimates technical and infrastructural barriers to participation and would benefit from more concrete pathways for citizen influence (e.g., funding mechanisms, deliberative processes). We agree that mapping barriers and comparing different concrete strategies is valuable, but a rigorous analysis thereof, plus a defense of specific strategies, would require detailed philosophical and (crucially) empirical analysis. This would imply a different methodology, different research question, and different conclusions than the one offered here, which would be beyond the scope of this position paper. Our contribution is more modest: to map and evaluate disagreement on democratizing AI as a concept and as a normative goal, not to defend one political strategy. That said, we recognize the value of concrete examples and now include references to bottom-up protests, citizen assemblies, consultative models, third-party auditing, and resource allocation.

3.R2 further asks us to clarify “what concrete mechanism or institutional structures [we] envision for enabling meaningful citizen redirection of AI *resources*.” We now specify, at the end of section 2.2, two broad options: (i) direct compensation of citizens for AI-related harms (citing case studies from the US and Italy), and (ii) civic decision-making about public AI funding, via mechanisms such as elections, ballot propositions, or citizen assemblies.

**3-5 Review Response3:**

gNzk

**3-6 Reaction To Review3:**

R3 offered an immensely helpful idea for framing our concrete examples more systematically: “Who is the intended audience for the recommendations (researchers, policymakers, industry)? What levers do they realistically have to act on this position?” We now comment on which levers different democratic actors can and should push (focusing primarily on the levers of citizens themselves in the context of the examples we offer, but also commenting briefly on policy-makers/elected officials and researchers). We focus comparatively less on the ethical and political obligations on tech industry practitioners because we think that current political and economic incentives in many jurisdictions make it hard (though perhaps not impossible) for the latter to significant boost citizens’ democratic *control* over AI. We briefly spell out this background assumption explicitly in the revised version of the paper.

Relatedly to R3's point, R1 raised an additional, partly 'out-of-scope' (according to R1) but insightful question: how to measure whether democratization genuinely redistributes power rather than producing “democracy-washing.” Addressing this fully would require normative *and* empirical research. Political scientists already use quantitative democracy indices (e.g. electoral integrity, polarization, proportionality, institutional fragility), but these measures face persistent critique (see Little & Meng 2024; Munck & Verkuilen 2002). We argue that quantitative approaches alone are insufficient and must be complemented with normative and qualitative input—a task we plan to pursue in future work. While developing a detailed comparative metric is beyond this paper’s scope, our argument does imply a coarse-grained acceptability threshold: democratization is insufficient if one actor (e.g. a tech company or bureaucracy) can unilaterally decide when and how to consult others, and can disregard those inputs at will. We briefly mention this example in the revised paper.

---

### Meta-Review · Area_Chair_Kscx · 2025-08-26

**Rating:** 7
**Confidence:** 4

**Strengths:**

Reviewers were generally positive about the paper. Specific strengths included:

- Overall, structure and organization was effective (with some room for improvement, see below)
- the "reframing" of this conversation (i.e., "democratizing AI") can provide an important contribution
- the paper attempts to engage with critics meaningfully, which could improve impact and achieve the "discussion goal" for this track

**Weaknesses:**

Overall, the main theme amongst reviewers was a desire for more concrete recommendations. Reviews suggested that readers will appreciate the abstract contributions of the work but will be left wanting more details about specific choices of mechanisms, specific institutions to support, etc.

Overall, reviewers seemed in favour of acceptance. It may be possible to add some additional concrete details with minor revisions, though a fully "ready to go version" of this paper may be a separate contribution entirely.

**Questions:**

Questions also echo the single (but important) theme in the weaknesses/areas for opportunities: what specific levers do people have? What specific mechanisms should we argue for, which institutions should we support, etc. What success stories should we look towards?

**Ethics:**

No concerns raised.

**Thoroughness:**

3

---

### Decision · Program_Chairs · 2025-09-26

Accept